# *Rotavirus alphagastroenteritidis*: Circulating Strains After the Introduction of the Rotavirus Vaccine (Rotarix^®^) in Luanda Province of Angola

**DOI:** 10.3390/v17060858

**Published:** 2025-06-17

**Authors:** Dikudila G. Vita, Cristina Santiso-Bellón, Manuel Lemos, Zoraima Neto, Elsa Fortes-Gabriel, Miguel Brito, Cruz S. Sebastião, Jesus Rodriguez-Diaz, Celso Cunha, Claudia Istrate

**Affiliations:** 1Faculty of Medicine, Agostinho Neto University, Luanda P.O. Box 116, Angola; dikudilavita@gmail.com (D.G.V.); manuel.lemos90@yahoo.com.br (M.L.); 2Department of Microbiology, School of Medicine, University of Valencia, Av. Blasco Ibáñez 15, 46010 Valencia, Spain; cristina.santiso@uv.es (C.S.-B.); jesus.rodriguez@uv.es (J.R.-D.); 3Instituto de Investigación INCLIVA, Hospital Clínico Universitario de Valencia, 46010 Valencia, Spain; 4National Health Research Institute (INIS), Maianga, Luanda P.O. Box 3635, Angola; zoraima.neto@gmail.com (Z.N.); cruzdossantos10@gmail.com (C.S.S.); 5Centro de Investigação em Saúde de Angola (CISA), Caxito, Luanda P.O. Box 5547, Angola; fortes.elsa@gmail.com (E.F.-G.); miguel.brito@estesl.ipl.pt (M.B.); 6Instituto Superior Técnico Militar (ISTM), Avenida Deolinda Rodrigues Km 9. Campo Militar do Grafanil, Luanda P.O. Box 5547, Angola; 7Health and Technology Research Center (H & TRC), Escola Superior de Tecnologia da Saúde de Lisboa, Instituto Politécnico de Lisboa, 1990-096 Lisbon, Portugal; 8Centro Nacional de Investigação Científica (CNIC), Avenida Ho Chi Minh, 201, Maianga, Luanda P.O. Box 3635, Angola; 9Global Health and Tropical Medicine (GHTM), Associate Laboratory in Translation and Innovation Towards Global Health, LA-REAL, Instituto de Higiene e Medicina Tropical (IHMT), Universidade NOVA de Lisboa, 1349-008 Lisbon, Portugal; 10Interdisciplinary Center for Research in Animal Health (CIISA), Faculty of Veterinary Medicine, University of Lisbon, 1300-477 Lisbon, Portugal; 11Associate Laboratory for Animal and Veterinary Sciences (AL4AnimalS), 1300-477 Lisbon, Portugal

**Keywords:** *rotavirus alphagastroenteritidis*, genotyping, Rotarix^®^, Luanda, Angola

## Abstract

*Rotavirus alphagastroenteritidis* (*R*. *alphagastroenteritidis*) remains the leading cause of pediatric diarrhea. Although Angola introduced Rotarix^®^, the human monovalent *R*. *alphagastroenteritidis* vaccine since 2014 as part of its routine childhood immunization program, no follow-up study has been conducted. The aim of this study was to evaluate the distribution of *R. alphagastroenteritidis* genotypes among children under five years of age, hospitalized with acute gastroenteritis (AGE), after the introduction of the rotavirus vaccine. To achieve this goal, stool samples collected between 2021 and 2022 from children under 5 years of age diagnosed with AGE at six hospitals in Luanda Province were analyzed. The *R. alphagastroenteritidis*-antigen immunochromatographic test (SD Bioline™, Abbott, Chicago, IL, USA) was performed, and 121 positive samples were genotyped. Ten samples were randomly selected for further Sanger sequencing. The results showed that the G9P[6] was the most prevalent genotype (17.3%), followed by G9P[8] (16.5%), G2P[4] (14.9%), G3P[6] (13.2%), G8P[6] (11.5%), and less frequently G12P[8] (9.1%), G1P[6] (4.1%), and G1P[8] (2.5%). The genotype combinations G3P[6], G8P[6], and G12P[8] were detected for the first time in Luanda Province. In conclusion, the emergence of new genotype combinations supports the need for continuous surveillance to identify the trend in *R. alphagastroenteritidis* infection and the emergence of new strains circulating in Luanda Province in the post-vaccination period.

## 1. Introduction

*Rotavirus alphagastroenteritidis* (*R. alphagastroenteritidis*) is the main causative agent of acute gastroenteritis (AGE) in infants and young children under five years of age and remains the most significant pathogen associated with infant mortality in most low- and middle-income countries (LMICs). It is estimated that 30–50% of childhood diarrheal hospitalizations in LMICs are due to *R. alphagastroenteritidis* infections [1,2]. In 2016, over 128,000 deaths were attributed to *R. alphagastroenteritidis* infections in children under five years of age [3], and about 90% of these fatalities occurred in LMICs, probably due to limited access to health services, lack of hydration therapy, and malnutrition [1]. In 2016, Angola was among the ten LMICs (India, Pakistan, Kenya, Democratic Republic of Congo, Niger, Angola, Ethiopia, Afghanistan, Nigeria, and Chad) with the highest diarrheal burden due to *R. alphagastroenteritidis*, and approximately 100 per 100,000 children died before reaching the age of five [4].

Since 2006, several *R. alphagastroenteritidis* oral live-attenuated vaccines have been licensed and prequalified by WHO for global use. Since then, *R. alphagastroenteritidis* vaccination has been introduced in the childhood immunization programs of more than 120 countries worldwide [5]. In this context, the first two approved *R. alphagastroenteritidis* vaccines were RotaTeq^®^ (RV5; Merck & Co. Inc., Whitehouse Station, NJ, USA) and Rotarix^®^ (RV1; GlaxoSmithKline Biologicals, Rixensart, Belgium). RotaTeq^®^ is a live pentavalent vaccine consisting of a mixture of mono-rearrangements of human and bovine *R. alphagastroenteritidis*, carrying genes encoding for human *R. alphagastroenteritidis* proteins G1, G2, G3, G4, and P[8] inserted into the bovine *R. alphagastroenteritidis* G6P[5] strain, while Rotarix^®^ is a monovalent vaccine derived from a human isolate G1P[8] [6]. The latter was introduced in Angola in 2014. The efficacy/effectiveness of the two vaccines was monitored over the years in both high-income countries and LMICs. Differences between countries have been reported due to various factors, such as malnutrition, breastfeeding, enteric pathogens, histo-blood group antigens, microbiota dysbiosis, and environmental enteropathy [7,8].

Since 2018, two additional vaccines were prequalified by WHO: Rotavac (Bharat Biotech International Ltd., Hyderabad, India) and Rotasiil (Serum Institute of India, Pune, India). Two additional vaccines are used only nationwide in China (Lanzhou lamb *R. alphagastroenteritidis*, manufactured by the Lanzhou Institute of Biomedical Products), and Vietnam (Rotavin-M1, manufactured by Polyvac, Hanoi, Vietnam) (https://www.who.int/teams/immunization-vaccines-and-biologicals/policies/position-papers/rotavirus (accessed on 20 March 2025).

*Rotavirus* belongs to the *Sedoreoviridae* family and is a large, non-enveloped virus with a triple-layered capsid and a diameter of up to 100 nm. The outer layer contains two capsid proteins, VP4 and VP7, the middle layer consists of the VP6 protein, while the inner core includes the VP2 protein associated with VP1 (RNA-dependent RNA polymerase) and VP3 (viral capping enzyme). The viral genome consists of 11 linear double-stranded RNA (dsRNA) segments, which are packaged entirely in the inner core layer [9]. The genome encodes six structural proteins (VP1-VP4, VP6, and VP7) and six non-structural proteins (NSP1-NSP6). The structural proteins integrated into the virion determine host specificity and mediate entry into the cell. Non-structural proteins, synthesized during infection, are involved in viral replication, pathogenesis, and inhibition of the host innate immune response, and also include the viral enterotoxin (NSP4) [1].

According to the antigenicity of the VP6 intermediate layer, at least ten different rotavirus species have been identified. Among them, *Rotavirus alphagastroenteritidis* (*R. alphagastroenteritidis*), *R. betagastroenteritidis*, *R. tritogastroenteritidis,* and *R. aspergastroenteritidis* infect both humans and other mammalian species [10].

Based on the antigenic and sequence variations of the two outer capsid proteins, VP7 and VP4, the *R. alphagastroenteritidis* strains were categorized into G (glycosylated) and P (protease-sensitive) genotypes, respectively, using a dual classification approach [11]. Although 42 G genotypes and 58 P genotypes have been described in humans and animals, only a few combinations of G and P genotypes are predominantly detected in humans [12]. The most frequently detected human rotavirus genotypes are G1P[8], G2P[4], G3P[8], G4P[8], G8P[8], G9P[8], and G12P[8] [12]. G1P[8] predominates in North America, Europe, and Australia, accounting for more than 70% of *R. alphagastroenteritidis* infections, and is less frequently detected in South America (30%), Asia (30%), and Africa (23%) [13].

The genetic diversity of *R. alphagastroenteritidis* is driven by interspecies transmission and gene rearrangement events, which represent important mechanisms of rotavirus evolution [14]. Humans can be infected by rotaviruses of animal origin, either through direct transmission or through strains resulting from the exchange of one or more genome segments between human and animal rotaviruses [14]. Several evolutionary studies have shown that human rotavirus strains originated from animal strains of independent ancestry. The evolution and diversity of *R. alphagastroenteritidis* strains are thought to be driven by various events that can occur in genome segments, including point mutations and genetic rearrangements through interspecies transmission [15]. The impact of these alterations on the effectiveness of the vaccine has yet to be determined [16].

This study is part of a larger investigation regarding rotavirus infection among children admitted with AGE in several public hospitals in Luanda province, Angola [17], and aims to provide baseline information on *R. alphagastroenteritidis* infection after the introduction of the Rotarix^®^ vaccine in 2014.

## 2. Materials and Methods

### 2.1. Study Design

A cross-sectional hospital-based study was conducted from April 2021 to May 2022 at the pediatric emergency and inpatient services of six public hospitals in Luanda Province, namely Luanda General Hospital (H. Luanda), General Hospital Cajueiros of Cazenga (H. Cajueiros of Cazenga), General Hospital of Kilamba Kiaxi (H. Kilamba Kiaxi), Municipal Hospital of Talatona (H. Talatona), Municipal Hospital of Zango (H. Zango), and Municipal Hospital Cacuaco (H. Cacuaco) [17]. This study involved 1251 children under five years of age, hospitalized with AGE. The primary inclusion criteria for participants were defined by diarrhea (≥3 episodes of loose or liquid stools per 24 h) and/or vomiting during a maximum of 7 days.

### 2.2. Sample Collection and R. alphagastroenteritidis Detection

Fecal samples were collected in sterile containers and screened locally for the presence of *R. alphagastroenteritidis* antigens using the WHO-approved SD Bioline^TM^ Rotavirus immunochromatographic rapid test (Abbott, Chicago, IL, USA), following the manufacturer’s instructions. Samples were stored on site at 4 °C until they were transported on ice packs to the National Institute for Health Research (INIS) in Luanda, Angola, where genotyping was performed.

### 2.3. Genotyping of R. alphagastroenteritidis Strains

Viral RNA was extracted from stool suspensions at 10% (*w*/*v*) using the NZY Viral RNA Isolation Kit (NZYTech, Lisbon, Portugal) according to the manufacturer’s instructions. The RNA was diluted in RNase-free water (60 µL) and stored at −20 °C until further analyses. Synthesis of cDNA was performed by reverse transcription (RT) with random hexamers, using a commercial kit (NZY First-strand cDNA Synthesis Kit, NZYTech, Lisbon, Portugal), according to the manufacturer’s instructions. *R. alphagastroenteritidis* G and P genotyping was performed using hemi-nested type specific multiplex PCRs, optimized to detect eight G-types (G1, G2, G3, G4, G8, G9, G10, and G12) and six P-types (P[4], P[6], P[8], P[9], P[10], and P[11]), as described previously [18,19]. The G and P genotypes were assigned according to the amplicon size, as visualized under ultraviolet light after electrophoresis on 2% (*w*/*v*) agarose gels, stained with GreenSafe Premium (NZYTech, Lisbon, Portugal).

### 2.4. Sequencing of R. alphagastroenteritidis Strains

Ten fecal samples were randomly selected for Sanger sequencing of both VP7 and VP4 gene amplicons, with the aim of covering the range of detected G and P genotypes, including non-typable strains. DNA sequencing was performed by STABVIDA Laboratories (Costa da Caparica, Portugal) using the corresponding first-round PCR primers. Complete sequences for both VP4 and VP7 genes were successfully obtained from 4 samples. In 3 samples, only the VP4 gene could be sequenced, while in the remaining 3, only the VP7 gene was recovered. These differences were due to the variability in PCR amplification efficiency and sequence quality among the samples.

### 2.5. Phylogenetic Analysis

The quality of the sequences was manually reviewed and adjusted using BioEdit v7.2.5 software [20]. All sequences were deposited in GenBank (NCBI) under accession numbers PQ139230 to PQ139236 (VP4 gene sequences) and PQ139237 to PQ139243 (VP7 gene sequences).

Phylogenetic trees were constructed using sequences from this study, along with reference sequences retrieved from GenBank. All sequences were aligned using Clustal X 2.1 [21] and matched with GeneDoc 2.7.000 [22]. The software used for phylogenetic tree construction and analysis was MEGA 7.0.23 [23].

The nucleotide substitution model for each tree was selected based on the lowest Bayesian Information Criterion (BIC) score and was also used to calculate the pairwise identity percentages. For the VP4 phylogenetic tree, the Hasegawa–Kishino–Yano model was selected, while the Tamura 3-parameter model was used for the VP7 tree. The evolutionary history was inferred using the Maximum Likelihood method [24] with a bootstrap test of 1000 replicates to assess reliability. To further assess genetic variability, nucleotide distance matrices for each genome segment were generated in MEGA 7.0.23 using the p-distance model, a built-in tool of the software.

Additionally, we analyzed differences in the antigenic regions of VP4 and VP7 [25,26,27,28] based on the deduced amino acid sequences of the Luanda strains, the Rotarix^®^ and RotaTeq^®^ vaccine strains, and other reference sequences included in the phylogenetic analysis.

### 2.6. Statistical Analysis

Descriptive statistics were used to assess the characteristics of the children enrolled in this study. Data were summarized by number, and absolute (n) and relative (%) frequencies of categorical variables were calculated. To investigate a possible association between age or severity of disease and genotypes (G or P), the children were divided into four age groups (0–6, 7–12, 13–24, >24 months) and 3 AGE groups according to the Vesikari scoring system (mild, moderate, severe), respectively. The association between proportions was assessed with a chi-squared (χ2) test of independence. A *p*-value < 0.05 was set as the threshold to assess statistical significance. Data were analyzed using software SPSS version 22 (IBM Corp, Armonk, NY, USA).

### 2.7. Ethical Considerations

This study was approved by the Independent Ethics Committee of the Faculty of Medicine, Agostinho Neto University, Luanda, Angola (Reference n° 12/2021, 15 January 2021). The parents or legal guardians of each child recruited for the study were informed of the study’s objectives and that participation was voluntary. Children were enrolled in the study after obtaining the informed consent form signed by parents or legal guardians.

## 3. Results

### 3.1. High Diversity of R. alphagastroenteritidis Strains Detected in Children Hospitalized with AGE in Luanda Province Public Hospitals

Stool samples from 1251 hospitalized children with AGE were screened for *R. alphagastroenteritidis* using an immunochromatographic rapid test. The overall *R. alphagastroenteritidis* detection rate was 57.8% (723/1251) [17]. One hundred and twenty-one *R. alphagastroenteritidis*-positive samples were genotyped, and sequencing confirmation was obtained for ten randomly selected samples, yielding a total of 14 sequences. The most prevalent G genotype was G9 (38.8%), followed by G2 (14.9%), G3 (14.9%), G8 (12.4%), and G12 (9.1%), while the most prevalent P genotypes were P[6] (47.1%), P[8] (28.1%), and P[4] (14.9%) (Table 1).

The relative frequency of G and P genotypes during the study period is shown in Figure 1. It can be observed that during the study period (2021–2022), G9P[6] (17.4%) was the most frequently found combination. However, G9P[8] (16.5%), G2P[4] (14.9%), G3P[6] (13.2%), G8P[6] (11.6%), and G12P[8] (9.1%) were also commonly detected. It is worth noting that strains such as G1P[6] (4.1%) and G1P[8] (2.5%), predominantly detected in a previous study from Angola [29] were now less frequently found. Various combinations of G or P non-typable strains were identified with frequencies below 2%.

The analysis of frequencies of *R. alphagastroenteritidis* genotypes in each of the six hospitals enrolled in the study showed some tendencies (Table 2). Genotypes G2P[4] and G9P[8] accounted for 68.7% (11/16) of all genotypes detected in H. Cacuaco, G9P[6] was the most frequent genotype in H. Luanda (34.5%; 10/29), G2P[4] and G9P[8] were the most prevalent in H. Zango (51.6%; 16/31), G12P[8] in H. Talatona (41.1%; 7/17), and the G3P[6] and G9P[8] genotypes in H. Kilamba Kiaxi (77.6%; 14/18).

### 3.2. Genotype Diversity, Age, and Severity of Disease

We investigated a possible relationship between genotypes and age or severity of disease. A large majority of genotyped samples (95.8%) belonged to children up to 12 months of age, a proportion similar to that found in the whole set of *R. alphagastroenteritidis*-positive cases (98.4%; 712/723) (Table 3). Among G genotypes, G9 was the most frequently detected in children less than 1 year old (35.5%), followed by G2 (14.8%), G3 (14.0%), G8 (12.3%), and G12 (9.0%). In addition, the two most frequent P genotypes in this age group, P[6] and P[8], accounted for 75.2% of all analyzed cases (42.9% P[6] and 32.3% P[8]). A chi-square test of independence showed that there is no significant association between the age group and *R. alphagastroenteritidis* genotypes.

According to the Vesikari scoring system, 90.0% children had severe *R. alphagastroenteritidis* (109/121) (Table 4). In this group, G9 was the predominant G genotype (36.6%), followed by G3 (17.4%), G2 (16.5%), G8 (11.9%), G12 (9.2%), and G1 (7.3%). However, the observed frequency differences were not statistically significant (*p* = 0.14). Moreover, two P genotypes, P[6] (47.1%) and P[8] (28.0%), were found to account for about three quarters of all severe *R. alphagastroenteritidis* cases. The third most frequent P genotype was P[4] (14.9%). A chi-square test of independence was performed, and a statistically significant association was found between the P genotype and severity of disease (*p* < 0.001). However, this association may reflect the predominance of P[6] and P[8] genotypes *in R. alphagastroenteritidis*-infected children, rather than a specific relation to severe diarrhea cases.

### 3.3. Phylogenetic Analysis of R. alphagastroenteritidis Strains Identified in Luanda Province Post-Vaccine Introduction

Despite being based on a limited number of sequences, the *R. alphagastroenteritidis* strains analyzed in Luanda Province showed low diversity, clustering within the same lineages of the respective genotypes for both VP4 and VP7.

The phylogenetic analysis of the *R. alphagastroenteritidis* VP4 gene (Figure 2) reveals that the Luanda Province sequences analyzed in this study form monophyletic clusters belonging to lineage III of genotype P[8] and lineage Ia of genotype P[6]. For both genotypes, all sequences included in this study share between 99.8% and 100% identity (Appendix A).

Regarding genotype P[8], the results indicate that sequences within lineage III exhibit identity percentages ranging from 79.9% (lineage IV) to 94.7% (lineage II) when compared to reference sequences from other lineages. Notably, lineage III includes sequence CSP10 (KT225724), which was circulating in Angola in 2013. This sequence shares substantial but not complete identity (95.6–95.9%) with the 2021–2022 sequences from the same region (DV66HMZ, DV105HMT, and DV117HMT) [29]. Additionally, these sequences display 92.1–92.3% and 86.4–86.7% identity with the vaccine strains RotaTeq^®^ and Rotarix^®^ (the vaccine administered in Angola), respectively.

For genotype P[6], the phylogenetic analysis shows that sequences DV68HKK, DV114HKK, DV120HKK, and DV177HKK cluster with a strain that was circulating in the region in 2012 (C27, KT225683), sharing 97.5% identity [29]. In contrast, identity analysis of the 2021 Luanda Province sequences with other lineages indicates a significant genetic divergence from subgroup Ib/c (86.6%) and even greater divergence from reference sequences of other lineages, with identity percentages ranging from 54.3% (lineage II) to 73.9% (lineage IV) (Appendix A).

The VP7 gene dendrogram (Figure 3) illustrates the clustering of Luanda Province sequences into different genotypes (G3, G8, G9, and G12). The 2021–2022 Luanda Province sequences belonging to genotype G3 (DV68HKK, DV177HKK, and DV114HKK) share 99.1–99.7% identity and cluster within lineage I alongside the RotaTeq^®^ vaccine strain, although with a high but incomplete identity (90.9–92.2%). The 2022 Luanda Province G8 sequences (DV34HECJ and DV540HGL) are identical to each other and exhibit high similarity (98.2%) with sequence CSMI10 (KT225669), belonging to a strain that circulated in Angola in 2012 [29]. Regarding sequence DV117HMT, classified as genotype G12, it clusters within lineage II and shares 95.6% identity with sequence AH6 (KT225672) from lineage III, which corresponds to a strain that was also circulating in Angola in 2012 [29]. Lastly, the 2022 sequence DV72HMCC clusters within lineage VI of genotype G9, distinct from sequence C84 (KT225671), which belongs to a strain found in Angola in 2012 that clusters within lineage III [29].

### 3.4. Comparison of the Deduced Antigenic Region of VP4 (VP8 *) and VP7

Table 5 and Table 6 show the positions of the deduced amino acid sequences corresponding to the antigenic regions of VP4 (VP8 *: 8-1, 8-2, 8-3, and 8-4) and VP7 (7-1a, 7-1b, and 7-2), respectively. These tables provide a detailed overview of the amino acid composition of these regions for each analyzed sample.

Concerning the VP4 (VP8 *) epitope, we identified seven amino acids conserved across all strains (Table 5): D100, S188, T190, and L103 in region 8-1; T180 and A183 in region 8-2; and N132 in region 8-3. This homology represents 28% of the epitope (7/25 aa).

Within the same genotype, all Luanda Province sequences exhibited 100% similarity in the antigenic region. Furthermore, our results show that P[8] genotype sequences display greater epitope conservation compared to those of the P[6] genotype sequences (60% and 44%, respectively). Finally, sequences belonging to the same lineage displayed higher homology when compared to sequences assigned to other lineages.

Regarding the P[8] genotype, Luanda Province sequences differed by three amino acids compared to other reference sequences of the same lineage (lineage III), namely in positions 194, 113, and 89. Specifically, we observed a N194D substitution in comparison with strains GER126-08 (Germany, 2008, genotype G12P[8]) and E9779 (France, 2013, genotype G1P[8]); a N113D substitution was observed in comparison with strains CSP10 (Angola, 2013, genotype G1P[8]) and BA20142 (Brazil, 2011, genotype G12P[8]); and a T89N substitution was found in comparison with all other P[8] strains, including the Rotarix^®^ vaccine strain (lineage I of genotype P[8]).

Overall, Luanda P[8] strains exhibited 24% variability in the antigenic region compared to the Rotarix^®^ vaccine strain. Specifically, we identified the following amino acid substitutions: D150E (both acidic), G195N (both polar uncharged), N125S (both polar uncharged), R131S (acidic and polar uncharged, respectively), D135N (acidic and polar uncharged, respectively), and T89N (both polar uncharged).

For the P[6] genotype, Luanda Province strains differed by a single amino acid P114N (nonpolar and polar, respectively) compared to strain C27 (G1P[6] genotype), which was circulating in Angola in 2012. In comparison with the Rotarix^®^ vaccine strain, Luanda Province P[6] strains exhibited 60% variability in the antigenic region.

Concerning the VP7 epitope, we performed a comparison of the antigenic region among the deduced amino acid sequences of the protein, as shown in Table 6. The limited nucleotide sequence lengths obtained for VP7 in samples DV540HGL (411 nt) and DV34HECJ (409 nt) did not allow for analysis of the corresponding complete antigenic regions. Additionally, sequences for the G1P[8] genotype in Luanda Province could not be obtained for comparison with the Rotarix^®^ vaccine strain.

Overall, the results show 20.7% homology in the VP7 epitope across all sequences analyzed, including amino acids T91, W98, Q104, and K291 from region 7-1a; Q201 from region 7-1b; and S190 and G264 from region 7-2.

Sequences from the same genotype exhibit slightly higher, though not complete, homology (82.8% for the G12 and G9 genotypes and 79.3% for the G8 genotype). The lowest homology was observed among G3 genotype sequences (62.1%).

Considering the number of missing positions due to nucleotide sequence length limitations, the variability in the VP7 epitope between the Luanda Province sequences and the Rotarix^®^ vaccine strain was 60.7% (G12 strains), 48.3% (G3 strains), 45.5% (G8 strains), and 53.6% (G9 strains).

On the other hand, the sample DV117HMT (G12P[8] genotype), found in Luanda Province in 2021–2022, showed no changes in the antigenic region compared to strain AH6, which circulated in Angola in 2012. Again, considering the absence of amino acid data from the Luanda Province G8 genotype strains (DV540HGL and DV34HECJ), which were found in 2021–2022, a comparison with strain CSMI10 (Angola 2012, G8P[6] genotype), belonging to the same lineage (lineage V), showed 100% homology (comparison of 10 positions in the antigenic region). Finally, the sample DV72HMCC (lineage VI of the G9P[ND] genotype), found in Luanda Province in 2021–2022, presented a single amino acid change, N100D (polar uncharged to acidic) compared to sample C84 (Angola 2012, the G9P[6] genotype).

## 4. Discussion

This study provides a significant update on the molecular epidemiology of *R. alphagastroenteritidis* circulating strains among children under five years of age, admitted with AGE at pediatric emergency services after the introduction of Rotarix^®^ vaccine in 2014. Our results showed a wide diversity of genotype combinations, a major shift in the distribution of genotypes during the study period, and a different dynamic of strains after the introduction of Rotarix^®^ into the Angolan Expanded Program for Immunization.

Some of the identified *R. alphagastroenteritidis* genotype combinations, G1P[8], G2P[4], G9P[8], and G12P[8], have been previously reported as among the six most common strains detected in humans globally (i.e., G1P[8], G2P[4], G3P[8], G4P[8], G9P[8], and G12P[8]) [30]. However, some uncommon genotypes thought to gain a growing epidemiological relevance in Africa [31], such as G8P[6], were detected with non-negligible frequencies (11.6%). Moreover, some genotypes detected here, such as G3P[6] (13.2%) and G9P[6] (17.6%), have been less commonly reported in Africa to date [31,32].

In our previous studies regarding the molecular epidemiology of *R. alphagastroenteritidis* in four provinces of Angola in 2011–2012 [29] and in the Northwestern Bengo Province in 2012–2013 [33], genotype G1P[8], which is also part of the vaccine makeup currently administered in Angola, was the most often detected, with frequencies of 50% and 47.2%, respectively. Nonetheless, in the present study, the frequency of detection of this genotype was strikingly low (2.5%). It is likely that the introduction of the vaccine or natural temporal fluctuations created selective pressure, driving the observed change in the circulating genotypes, a phenomenon reported in other geographic regions. This drastic reduction in the circulation of the G1P[8] strain with concomitant replacement by novel emerging *R. alphagastroenteritidis* strains, such as G3P[8], was earlier reported in Italy, in 2018–2019, after the introduction of the Rotarix^®^ vaccine [34]. In the two above-mentioned studies from Angola [29,33], genotype G1P[6] was the second most prevalent, with frequencies around 29%. Herein, the detection rate of this genotype dramatically decreased to 4.1%. An opposite tendency was observed with genotype G9P[6], which was among the less commonly detected genotypes in 2012 (1.7%) [29] and has become the most prevalent genotype in Luanda Province after the introduction of the Rotarix^®^ vaccine, with a frequency of 17.3%. Notably, the genotype G3P[6], not identified in previous studies in Angola, was now among the most commonly detected genotypes (13.2%). It is important to note that, similarly to our data, earlier studies from Mozambique [35] and Malawi [36] also reported an emergence of G3 strains after the introduction of the Rotarix^®^ vaccine.

We detected six G-types of *R. alphagastroenteritidis* (G1, G2, G3, G8, G9, and G12), with G9 being the most frequent genotype in all hospitals (38.8%). Overall, a shift was observed from globally common G-types (G1–G4, G9, G12) to G9, G3, G8, and G12. The decline in the G1-type (predominant in the pre-vaccination era) and the dominance of G9 was also acknowledged in other African countries, such as South Africa [30]. As for the P-type, P[6] was recorded in high proportion at all study sites, followed by P[8] and P[4]. Genotype P[6] was now found as the most frequent (47.1%), although P[8] and P[4] were also observed with a significant frequency (28% and 14.9%, respectively). The three P genotypes, P[4], P[6], and P[8] accounted for about 90% of all circulating *R. alphagastroenteritidis* strains detected after introduction of the Rotarix^®^ vaccine.

When comparing the most prevalent G/P combinations before and after the introduction of Rotarix^®^, G9P[6] was the most prevalent genotype combination in this postvaccination study period, while G1P[8] was the most prevalent genotype combination in the pre-vaccination period [29,33]. The genotype combination G9P[6] can be detected worldwide, namely in countries and regions where the vaccine has been introduced [31,37]. However, a different tendency was reported in early epidemiological studies performed in India [38]. These studies showed high detection rates of G9P[6], leading to the inclusion of this genotype in the development of the first Indian rotavirus vaccine.

The reduced detection rate of the G1P[8] genotype combination may be attributed to the introduction of the vaccine, since Rotarix^®^ is a monovalent vaccine specific to this genotype. Similarly to Angola, other countries that have implemented vaccination programs with the Rotarix^®^ vaccine also reported a decline in the circulation of G1P[8] genotype combinations. This post-vaccination tendency can be illustrated by the reduction in G1P[8] frequencies and the simultaneous increase in the detection rate of other non-G1P[8] combinations in South Africa [39], as well as the rise in heterotypic strains, such as G2P[4] in Brasil [40].

In the VP4 (VP8 *) epitope, seven out of 25 amino acids (D100, L103, N132, T180, A183, S188, and T190) were found to be conserved in all strains (28% homology). Both P[6] and P[8] genotypes displayed high homology with a strain circulating in Angola in 2012, as well as with other reference genotypes. They differed only by 1 and 3 amino acids, respectively, in the VP4 (VP8 *) epitope. In addition, further comparison of the P[6] and P[8] epitopes of the Luanda Province strains with the Rotarix^®^ vaccine strain showed 60% and 24%, respectively, variability in the antigenic region.

With regard to the VP7 epitope, we could not obtain sequences for the G1P[8] genotype, and for two samples, DV540HGL and DV34HECJ, the length of the obtained sequence was limited (409 nt and 411 nt, respectively). Overall, the analysis of the deduced amino acid sequences showed 20.7% homology in the VP7 epitope. Furthermore, when compared to the Rotarix^®^ vaccine strain, the variability in the VP7 epitope displayed by Luanda Province sequences was 48.3% (G3 strains), 45.5% (G8 strains), 53.6% (G9 strains), and 60.7% (G12 strains).

Given the limited number of sequences analyzed, any interpretation of amino acid changes remains speculative. Nonetheless, substitutions involving residues with differing chemical properties—such as nonpolar, polar, and charged amino acids—in key regions of VP4 (VP8) and VP7 were observed, which may suggest potential conformational alterations in antigenic epitopes. However, the biological relevance of these putative changes is uncertain and requires confirmation through analysis of a larger dataset.

The detected changes in amino acids in the epitope regions of circulating strains, when compared to those of the vaccine strain, may also be indicative of the presence of a selective immune pressure. Although speculative, this pressure could accelerate the emergence of new *R. alphagastroenteritidis* strains capable of evading the immune response. Monitoring this emergence in the post-vaccination era is a task of utmost relevance to reduce the burden of *R. alphagastroenteritidis* infection, especially in LMIC.

This study has some limitations that should be considered. First, because it was a cross-sectional study, it was not possible to establish causal relationships between the detected genotypes and clinical severity. Second, our sampling was carried out only in public hospitals in the province of Luanda, which may not reflect the situation in other regions of Angola. Third, genetic analysis was performed on a limited number of samples, which may not capture the full diversity of circulating strains. Finally, the use of rapid tests for the initial detection of *R. alphagastroenteritidis* may have reduced sensitivity, especially in cases with low viral load. Nevertheless, our findings are essential for evaluating the effectiveness of childhood immunization against rotavirus, as well as guiding public policies for vaccination and control of gastroenteritis in Angola.

## 5. Conclusions

This is the first report to describe the circulation of *R. alphagastroenteritidis* genotypes in Luanda Province, Angola, and revealed a change in the circulating genotypes after the introduction of the Rotarix^®^ vaccine in 2014. However, due to the short surveillance period, it is unclear whether the changes observed are due to the introduction of the vaccine or are a consequence of the natural variation of the strain. In addition, the emergence of unusual strains such as G9P[8], G3P[6], and G12P[8] has also been observed, which supports the need for continued genomic and epidemiological surveillance in Luanda to monitor changes due to possible vaccine pressure and, consequently, asses their effect on vaccine efficacy.

## Figures and Tables

**Figure 1 viruses-17-00858-f001:**
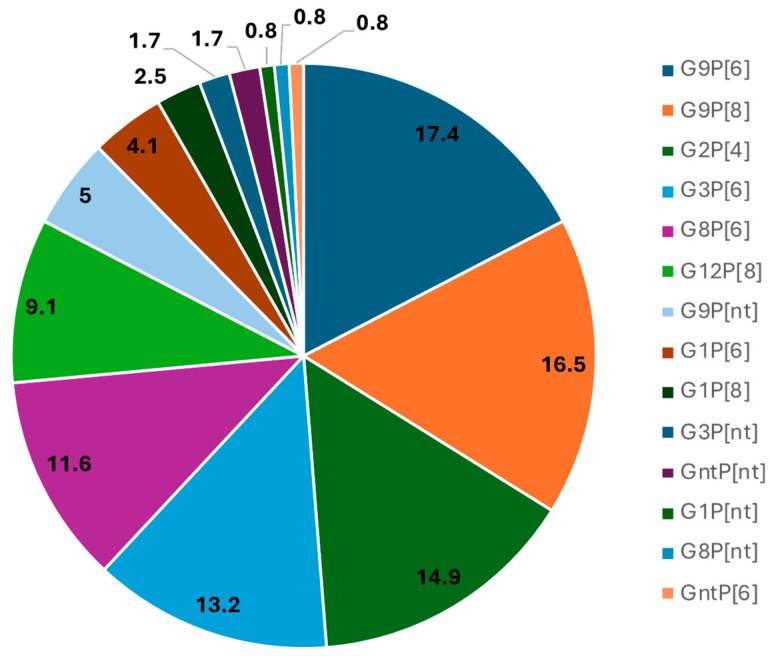
*R. alphagastroenteritidis* strain diversity in hospitals of Luanda Province (2021/2022). *R. alphagastroenteritidis*-positive samples were genotyped using a hemi-nested type-specific multiplex PCR protocol optimized to detect eight G-types (G1, G2, G3, G4, G8, G9, G10, and G11) and six P-types (P[4], P[6], P[8], P[9], P[10]). The results are presented as percentage of each G and P combination in relation to the total number of positive samples (n = 121).

**Figure 2 viruses-17-00858-f002:**
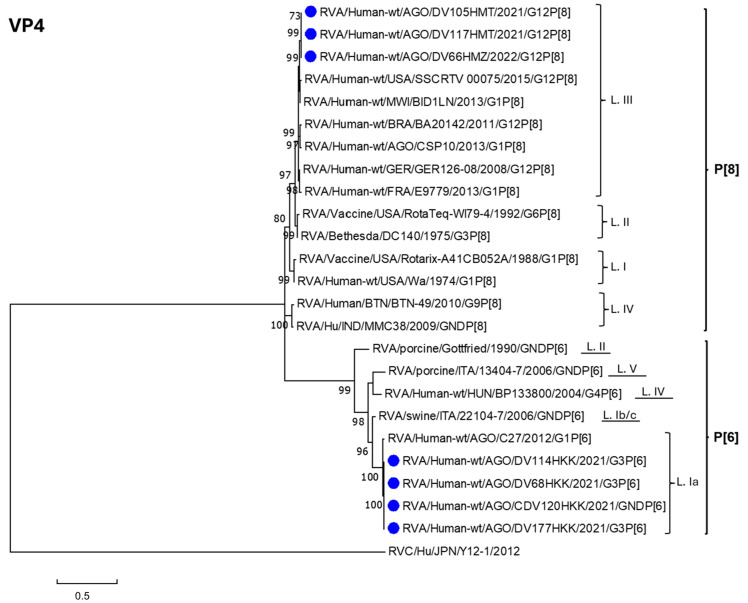
Molecular phylogenetic analysis by Maximum Likelihood (ML) method. The ML tree with the highest log likelihood (−3898.30) is shown. The percentage of trees in which the associated taxa clustered together is shown next to the branches. Initial tree(s) for the heuristic search were obtained automatically by applying neighbor joining and BioNJ algorithms to a matrix of pairwise distances estimated using the Maximum Composite Likelihood (MCL) approach, and then selecting the topology with superior log likelihood value. A discrete Gamma distribution was used to model evolutionary rate differences among sites (5 categories (+G, parameter = 2.0878)). The rate variation model allowed for some sites to be evolutionarily invariable ([+I], 12.52% sites). The tree is drawn to scale, with branch lengths measured in the number of substitutions per site. The analysis involved 25 nucleotide sequences. All positions containing gaps and missing data were removed. There was a total of 563 positions in the final dataset. Evolutionary analyses were conducted in MEGA7 [24]. RVA = *R. alphagastroenteritidis*. Blue dots indicate *R. alphagastroenteritidis* sequences obtained from patients’ samples and analyzed in this study.

**Figure 3 viruses-17-00858-f003:**
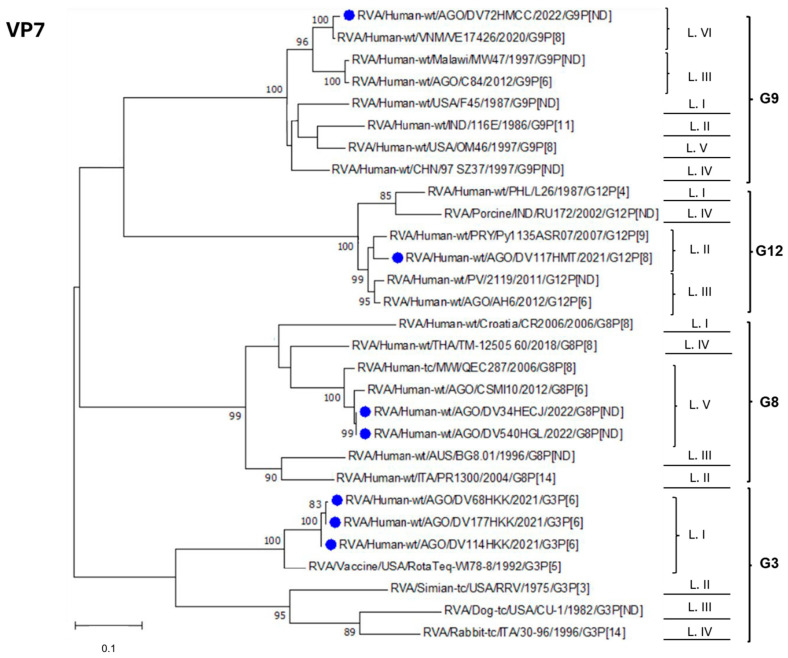
Molecular Phylogenetic analysis by Maximum Likelihood (ML) method. The ML tree with the highest log likelihood (−2981.70) is shown. The percentage of trees in which the associated taxa clustered together is shown next to the branches. Initial tree(s) for the heuristic search were obtained automatically by applying neighbor joining and BioNJ algorithms to a matrix of pairwise distances estimated using the Maximum Composite Likelihood (MCL) approach, and then selecting the topology with superior log likelihood value. A discrete Gamma distribution was used to model evolutionary rate differences among sites (five categories (+G, parameter = 1.5436)). The rate variation model allowed for some sites to be evolutionarily invariable ([+I], 26.98% sites). The tree is drawn to scale, with branch lengths measured as the number of substitutions per site. The analysis involved 29 nucleotide sequences. All positions containing gaps and missing data were eliminated. There was a total of 354 positions in the final dataset. Evolutionary analyses were conducted in MEGA7 [23]. RVA = *R. alphagastroenteritidis.* Blue dots indicate *R. alphagastroenteritidis* sequences obtained from patients’ samples and analyzed in this study.

**Table 1 viruses-17-00858-t001:** Prevalence of G and P-types in hospitalized children with AGE during the study period (2021–2022).

**G-type**	**N**	**%**
G1	9	7.4
G2	18	14.9
G3	18	14.9
G8	15	12.4
G9	46	38.8
G12	11	9.1
Gnt ^a^	3	2.5
Total	121	100
**P-type**	**N**	**%**
P4	18	14.8
P6	57	47.1
P8	34	28.0
Pnt ^b^	12	9.9
Total	121	100

^a^ Strains that were non-typable for G; ^b^ strains that were non-typable for P.

**Table 2 viruses-17-00858-t002:** Genotypic combinations of *R. alphagastroenteritidis* strains circulating in six hospitals in Luanda Province (2021–2022).

Genotype Combination	Cacuaco n (%)	Luanda n (%)	Zango n (%)	Cajueiros n (%)	Talatona n (%)	K. Kiaxi n (%)	Total n (%)
G1P[6]	0 (0.0)	2 (6.9)	3 (9.7)	0 (0.0)	0 (00.0)	0 (0.0)	5 (4.1)
G1P[8]	0 (0.0)	0 (0.0)	0 (0.0)	0 (0.0)	3 (17.6)	0 (0.0)	3 (2.5)
G1Pnt *	0 (0.0)	1 (3.4)	0 (0.0)	0 (0.0)	0 (0.0)	0 (0.0)	1 (0.8)
G2P[4]	5 (31.2)	3 (10.3)	9 (29.0)	0 (0.0)	1 (5.9)	0 (0.0)	18 (14.9)
G3P[6]	0 (0.0)	2 (6.9)	1 (3.2)	1 (10)	2 (11.8)	10 (55.6)	16 (13.2)
G3Pnt *	0 (0.0)	0 (0.0)	1 (3.2)	0 (0.0)	0 (0.0)	1 (5.6)	2 (1.6)
G8P[6]	1 (6.2)	3 (10.3)	3 (9.7)	4 (40.0)	2 (11.8)	1 (5.6)	14 (11.5)
G8Pnt *	0 (0.0)	1 (3.4)	0 (0.0)	0 (0.0)	0 (0.0)	0 (0.0)	1 (1.7)
G9P[6]	3 (18.8)	10(34.5)	5 (16.1)	1 (10)	1 (5.9)	1 (5.6)	21 (17.3)
G9P[8]	6 (37.5)	3(10.3)	7 (22.6)	0 (0.0)	0 (0.0)	4 (22.0)	20 (16.5)
G9Pnt *	1 (6.3)	0 (0.0)	0 (0.0)	3 (30.0)	1 (5.9)	1 (5.6)	6 (4.9)
G12P[8]	0 (0.0)	2 (6.9)	2 (6.5)	0 (0.0)	7 (41.1)	0 (0.0)	11 (9.1)
* GntP[6]	0 (0.0)	0 (0.0)	0 (0.0)	1 (10)	0 (0.0)	0 (0.0)	1 (0.8)
* GntPnt	0 (0.0)	2 (6.9)	0 (0.0)	0 (0.0)	0 (0.0)	0 (0.0)	2 (1.7)
Total	16 (100)	29 (100)	31 (100)	10 (100)	17 (100)	18 (100)	121 (100)

* Strains that were non-typable for either G-type or P-type.

**Table 3 viruses-17-00858-t003:** Distribution of *R. alphagastroenteritidis* genotypes by age group.

Age Group (Months)					
Genotype	0–6 n (%)	7–12 n (%)	13–24 n (%)	>24 n (%)	Total
G1P[6]	3 (3.4)	2 (6.9)	0 (0.0)	0 (0.0)	5 (4.1)
G1P[8]	3 (3.4)	0 (0.0)	0 (0.0)	0 (0.0)	3 (2.5)
G1PN	1 (1.1)	0 (0.0)	0 (0.0)	0 (0.0)	1 (0.8)
G2P[4]	14 (16.1)	4 (13.8)	0 (0.0)	0 (0.0)	18 (14.9)
G3P[6]	12 (13.8)	3 (10.3)	0 (0.0)	1 (33.3)	16 (13.2)
G3PN	2 (2.3)	0 (0.0)	0 (0.0)	0 (0.0)	2 (1.6)
G8P[6]	11 (12.6)	3 (10.3)	0 (0.0)	0 (0.0)	14 (11.5)
G8PN	1 (1.1)	0 (0.0)	0 (0.0)	0 (0.0)	1 (1.7)
G9P[6]	9 (10.3)	8 (27.5)	2 (100)	2 (66.7)	21 (17.3)
G9P[8]	14 (16.1)	6 (20.7)	0 (0.0)	0 (0.0)	20 (16.5)
G9PN	4 (4.6)	2 (6.9)	0 (0.0)	0 (0.0)	6 (4.9)
G12P[8]	10 (11.5)	1 (3.4)	0 (0.0)	0 (0.0)	11 (9.1)
* GntP[6]	1 (1.1)	0 (0.0)	0 (0.0)	0 (0.0)	1 (0.8)
* GntPN	2 (2.9)	0 (0.0)	0 (0.0)	0 (0.0)	2 (1.7)
Total	87 (100)	29 (100)	2 (100)	3 (100)	121 (100)

* Strains that were non-typable for G.

**Table 4 viruses-17-00858-t004:** Distribution of genotype according to the severity of *R. alphagastroenteritidis* AGE using the Vesikari scoring system.

Severity of Diarrhea				
Genotype	Mild (<7) n (%)	Moderate (7–10) n (%)	Severe (≥11) n (%)	Total n (%)
G1P[6]	0 (0.0)	0 (0.0)	5 (4.5)	5 (4.1)
G1P[8]	0 (0.0)	1 (10.0)	2 (1.8)	3 (2.5)
G1Pnt	0 (0.0)	0 (0.0)	1 (0.9)	1 (0.8)
G2P[4]	0 (0.0)	0 (0.0)	18 (16.5)	18 (14.9)
G3P[6]	0 (0.0)	1 (10.0)	15 (15.6)	16 (13.2)
G3PN	0 (0.0)	0 (0.0)	2 (1.8)	2 (1.6)
G8P[6]	0 (0.0)	2 (20.0)	12 (11.0)	14 (11.5)
G8Pnt	0 (0.0)	0 (0.0)	1 (0.9)	1 (1.7)
G9P[6]	2 (100)	1 (10.0)	18 (16.5)	21 (17.3)
G9P[8]	0 (0.0)	2 (20.0)	18 (16.5)	20 (16.5)
G9PN	0 (0.0)	2 (20.0)	4 (3.6)	6 (4.9)
G12P[8]	0 (0.0)	1 (10.0)	10 (9.2)	11 (9.1)
* GntP[6]	0 (0.0)	0 (0.0)	1 (0.9)	1 (0.8)
* GntPN	0 (0.0)	0 (0.0)	2 (1.8)	2 (1.7)
Total	2 (100)	10 (100)	109 (100)	121 (100)

* Strains that were non-typable for either G-type or P-type.

**Table 5 viruses-17-00858-t005:** VP4 (VP8 *) antigenic regions (8-1, 8-2, 8-3, and 8-4), showing the amino acids present in each strain included in the study.

Lineage		Strain	VP4 (VP8 *) EPITOPE																											
			**8-1**												**8-2**			**8-3**										**8-4**		
			**100**	**146**	**148**	**150**	**188**	**190**	**192**	**193**	**194**	**195**	**196**		**180**	**183**		**113**	**114**	**115**	**116**	**125**	**131**	**132**	**133**	**135**		**87**	**88**	**89**
**P[8]**	**III**	**RVA/Hu/AGO/DV105HMT/2021/G12P[8]**	**D**	**S**	**Q**	**D**	**S**	**T**	**N**	**L**	**N**	**G**	**I**		**T**	**A**		**N**	**P**	**V**	**D**	**N**	**R**	**N**	**D**	**D**		**N**	**T**	**T**
		**RVA/Hu/AGO/DV117HMT/2021/G12P[8]**	**.**	**.**	**.**	**.**	**.**	**.**	**.**	**.**	**.**	**.**	**.**		**.**	**.**		**.**	**.**	**.**	**.**	**.**	**.**	**.**	**.**	**.**		**.**	**.**	**.**
		**RVA/Hu/AGO/DV66HMZ/2022/G12P[8]**	**.**	**.**	**.**	**.**	**.**	**.**	**.**	**.**	**.**	**.**	**.**		**.**	**.**		**.**	**.**	**.**	**.**	**.**	**.**	**.**	**.**	**.**		**.**	**.**	**.**
		RVA/Hu/MWI/BID1LN/2013/G1P[8]	**.**	**.**	**.**	**.**	**.**	**.**	**.**	**.**	**.**	**.**	**.**		**.**	**.**		**.**	**.**	**.**	**.**	**.**	**.**	**.**	**.**	**.**		**.**	**.**	**N**
		RVA/Hu/USA/SSCRTV 00075/2015/G12P[8]	**.**	**.**	**.**	**.**	**.**	**.**	**.**	**.**	**.**	**.**	**.**		**.**	**.**		**.**	**.**	**.**	**.**	**.**	**.**	**.**	**.**	**.**		**.**	**.**	**N**
		Hu/GER126-08/GER/2008/G12P[8]	**.**	**.**	**.**	**.**	**.**	**.**	**.**	**.**	**D**	**.**	**.**		**.**	**.**		**.**	**.**	**.**	**.**	**.**	**.**	**.**	**.**	**.**		**.**	**.**	**N**
		RVA/Hu/AGO/CSP10/2013/G1P[8]	**.**	**.**	**.**	**.**	**.**	**.**	**.**	**.**	**.**	**.**	**.**		**.**	**.**		**D**	**.**	**.**	**.**	**.**	**.**	**.**	**.**	**.**		**.**	**.**	**N**
		RVA/Hu/BRA/BA20142/2011/G12P[8]	**.**	**.**	**.**	**.**	**.**	**.**	**.**	**.**	**.**	**.**	**.**		**.**	**.**		**D**	**.**	**.**	**.**	**.**	**.**	**.**	**.**	**.**		**.**	**.**	**N**
		RVA/Hu/FRA/E9779/2013/G1P[8]	**.**	**.**	**.**	**.**	**.**	**.**	**.**	**.**	**D**	**.**	**.**		**.**	**.**		**.**	**.**	**.**	**.**	**.**	**.**	**.**	**.**	**.**		**.**	**.**	**N**
	**IV**	RVA/Hu/BTN-49/BTN/2010/G9P[8]	**.**	**.**	**.**	**E**	**.**	**.**	**D**	**.**	**T**	**S**	**.**		**.**	**.**		**D**	**.**	**.**	**.**	**S**	**.**	**.**	**.**	**N**		**.**	**.**	**N**
		RVA/Hu/IND/MMC38/2009/GNDP[8]	**.**	**G**	**.**	**E**	**.**	**.**	**D**	**.**	**T**	**S**	**.**		**.**	**.**		**D**	**.**	**.**	**.**	**S**	**.**	**.**	**.**	**N**		**.**	**.**	**N**
	**II**	Vac/RotaTeq-WI79-4/USA/1992/G6P[8]	**.**	**.**	**.**	**E**	**.**	**.**	**.**	**.**	**.**	**D**	**.**		**.**	**.**		**.**	**.**	**.**	**.**	**.**	**.**	**.**	**.**	**.**		**.**	**.**	**N**
		RVA/Bethesda/DC140/1975/G3P[8]	**.**	**.**	**.**	**E**	**.**	**.**	**.**	**.**	**.**	**D**	**.**		**.**	**.**		**.**	**.**	**.**	**.**	**.**	**.**	**.**	**.**	**.**		**.**	**.**	**N**
	**I**	Vac/Rotarix/USA/2009/G1P[8]	**.**	**.**	**.**	**E**	**.**	**.**	**.**	**.**	**.**	**N**	**.**		**.**	**.**		**.**	**.**	**.**	**.**	**S**	**S**	**.**	**.**	**N**		**.**	**.**	**N**
		RVA/Hu/USA/Wa/1974/G1P[8]	**.**	**.**	**.**	**E**	**.**	**.**	**.**	**.**	**.**	**N**	**.**		**.**	**.**		**.**	**.**	**.**	**.**	**S**	**S**	**.**	**.**	**N**		**.**	**.**	**N**
**P[6]**	**II**	RVA/porcine/Gottfried/1990/GNDP[6]	**.**	**N**	**N**	**D**	**.**	**.**	**.**	**.**	**P**	**D**	**V**		**.**	**.**		**P**	**S**	**Q**	**.**	**V**	**E**	**.**	**S**	**.**		**I**	**N**	**K**
	**Ia**	**RVA/Hu/AGO/DV114HKK/2021/G3P[6]**	**.**	**.**	**S**	**E**	**.**	**.**	**.**	**.**	**S**	**E**	**V**		**.**	**.**		**T**	**S**	**Q**	**S**	**T**	**E**	**.**	**N**	**S**		**T**	**N**	**Q**
		**RVA/Hu/AGO/DV68HKK/2021/G3P[6]**	**.**	**.**	**S**	**E**	**.**	**.**	**.**	**.**	**S**	**E**	**V**		**.**	**.**		**T**	**S**	**Q**	**S**	**T**	**E**	**.**	**N**	**S**		**T**	**N**	**Q**
		**RVA/Hu/AGO/DV177HKK/2021/G3P[6]**	**.**	**.**	**S**	**E**	**.**	**.**	**.**	**.**	**S**	**E**	**V**		**.**	**.**		**T**	**S**	**Q**	**S**	**T**	**E**	**.**	**N**	**S**		**T**	**N**	**Q**
		**RVA/Hu/AGO/CDV120HKK/2021/GNDP[6]**	**.**	**.**	**S**	**E**	**.**	**.**	**.**	**.**	**S**	**E**	**V**		**.**	**.**		**T**	**S**	**Q**	**S**	**T**	**E**	**.**	**N**	**S**		**T**	**N**	**Q**
		RVA/Hu/AGO/C27/2012/G1P[6]	**.**	**.**	**S**	**E**	**.**	**.**	**.**	**.**	**S**	**E**	**V**		**.**	**.**		**T**	**N**	**Q**	**S**	**T**	**E**	**.**	**N**	**S**		**T**	**N**	**Q**
	**Ib/c**	RVA/swine/ITA/22104-7/2006/GNDP[6]	**.**	**.**	**S**	**E**	**.**	**.**	**.**	**.**	**S**	**E**	**V**		**.**	**.**		**T**	**S**	**Q**	**S**	**T**	**E**	**.**	**N**	**N**		**T**	**N**	**Q**
	**IV**	RVA/Hu/HUN/BP133800/2004/G4P[6]	**.**	**N**	**S**	**E**	**.**	**.**	**.**	**.**	**S**	**E**	**.**		**.**	**.**		**T**	**N**	**Q**	**S**	**T**	**E**	**.**	**S**	**N**		**T**	**N**	**Q**
	**V**	RVA/porcine/ITA/13404-7/2006/GNDP[6]	**.**	**N**	**S**	**E**	**.**	**.**	**.**	**.**	**P**	**D**	**.**		**.**	**.**		**T**	**N**	**Q**	**S**	**M**	**E**	**.**	**N**	**S**		**T**	**N**	**Q**

RVA = *R. alphagastroenteritidis*.

**Table 6 viruses-17-00858-t006:** VP7 antigenic regions (7-1a, 7-1b, and 7-2), showing the amino acids present in each strain included in the study.

Lineage			Strain	VP7 EPITOPE																														
				7-1a															7-1b							7-2								
				87	91	94	96	97	98	99	100	104	123	125	129	130	291		201	211	212	213	238	242		143	145	146	147	148	190	217	221	264
**G12**		**II**	**RVA/Hu/AGO/DV117HMT/2021/G12P[8]**	**S**	**T**	**T**	**P**	**D**	**W**	**T**	**N**	**Q**	**D**	**S**	**V**	**D**	**…**		**Q**	**D**	**V**	**T**	**N**	**N**		**Q**	**Q**	**N**	**S**	**L**	**S**	**E**	**A**	**G**
			RVA/Hu/Py1135ASR07/PRY/2007/G12P[9]	**.**	**.**	**.**	**.**	**.**	**.**	**.**	**.**	**.**	**.**	**A**	**.**	**.**	**K**		**.**	**.**	**.**	**.**	**.**	**.**		**.**	**.**	**.**	**.**	**.**	**.**	**.**	**.**	**.**
		**III**	RVA/Hu/2119/PV/2011/G12P[ND]	**.**	**.**	**.**	**.**	**.**	**.**	**.**	**.**	**.**	**.**	**.**	**.**	**.**	**…**		**.**	**.**	**.**	**.**	**.**	**.**		**.**	**.**	**.**	**.**	**.**	**.**	**.**	**.**	**.**
			RVA/Hu/AGO/AH6/2012/G12P[6]	**.**	**.**	**.**	**.**	**.**	**.**	**.**	**.**	**.**	**.**	**.**	**.**	**.**	**…**		**.**	**.**	**.**	**.**	**.**	**.**		**.**	**.**	**.**	**.**	**.**	**.**	**.**	**.**	**.**
		**I**	Hu/L26/PHL/1987/G12P[4]	**N**	**.**	**.**	**.**	**.**	**.**	**.**	**H**	**.**	**.**	**A**	**.**	**N**	**K**		**.**	**.**	**.**	**A**	**.**	**.**		**.**	**.**	**.**	**.**	**.**	**.**	**.**	**.**	**.**
		**IV**	Po/RU172/IND/2002/G12PX	**N**	**.**	**.**	**.**	**.**	**.**	**.**	**.**	**.**	**.**	**A**	**.**	**.**	**K**		**.**	**.**	**.**	**A**	**.**	**.**		**.**	**.**	**.**	**.**	**.**	**.**	**.**	**.**	**.**
**G1**			**Vac/Rotarix/USA/2009/G1P[8]**	**T**	**.**	**N**	**G**	**E**	**.**	**K**	**D**	**.**	**S**	**V**	**.**	**.**	**K**		**.**	**N**	**.**	**D**	**.**	**T**		**K**	**D**	**Q**	**N**	**.**	**.**	**M**	**N**	**.**
**G3**	**I**		Vac/RotaTeq-W178-8/USA/1992/G3P[5]	**T**	**.**	**N**	**N**	**S**	**.**	**K**	**D**	**.**	**.**	**A**	**.**	**.**	**K**		**.**	**.**	**A**	**N**	**K**	**D**		**K**	**D**	**A**	**T**	**.**	**.**	**.**	**.**	**.**
			**RVA/Hu/AGO/DV114HKK/2021/G3P[6]**	**T**	**.**	**N**	**N**	**S**	**.**	**K**	**.**	**.**	**.**	**A**	**.**	**.**	**K**		**.**	**.**	**T**	**N**	**.**	**.**		**K**	**D**	**V**	**T**	**.**	**.**	**.**	**D**	**.**
			**RVA/Hu/AGO/DV177HKK/2021/G3P[6]**	**T**	**.**	**N**	**N**	**S**	**.**	**K**	**.**	**.**	**.**	**A**	**.**	**.**	**K**		**.**	**.**	**T**	**N**	**.**	**.**		**K**	**D**	**V**	**T**	**.**	**.**	**.**	**D**	**.**
			**RVA/Hu/AGO/DV68HKK/2021/G3P[6]**	**T**	**.**	**N**	**N**	**S**	**.**	**K**	**.**	**.**	**.**	**A**	**.**	**.**	**…**		**.**	**.**	**T**	**N**	**.**	**.**		**K**	**D**	**V**	**T**	**.**	**.**	**.**	**D**	**.**
	**II**		RVA/Simian/USA/RRV/1975/G3P[3]	**T**	**.**	**N**	**N**	**S**	**.**	**K**	**D**	**.**	**.**	**A**	**.**	**.**	**K**		**.**	**.**	**T**	**A**	**D**	**A**		**K**	**D**	**A**	**T**	**.**	**.**	**.**	**.**	**.**
	**III**		RVA/Dog/USA/CU-1/1982/G3P[ND]	**T**	**.**	**N**	**N**	**S**	**.**	**K**	**D**	**.**	**.**	**A**	**.**	**.**	**K**		**.**	**.**	**.**	**S**	**D**	**T**		**K**	**D**	**A**	**A**	**.**	**.**	**.**	**T**	**.**
	**IV**		RVA/Rabbit/ITA/30-96/1996/G3P[14]	**T**	**.**	**N**	**N**	**S**	**.**	**K**	**D**	**.**	**.**	**V**	**.**	**.**	**K**		**.**	**N**	**A**	**A**	**D**	**A**		**K**	**D**	**A**	**A**	**.**	**.**	**.**	**.**	**.**
**G8**	**V**		**RVA/Hu/AGO/DV540HGL/2022/G8P[ND]**	**…**	**…**	**…**	**…**	**…**	**…**	**…**	**…**	**…**	**…**	**…**	**…**	**…**	**K**		**.**	**.**	**T**	**.**	**.**	**T**		**…**	**…**	**…**	**…**	**…**	**.**	**.**	**.**	**.**
			**RVA/Hu/AGO/DV34HECJ/2022/G8P[ND]**	**…**	**…**	**…**	**…**	**…**	**…**	**…**	**…**	**…**	**…**	**…**	**…**	**…**	**K**		**.**	**.**	**T**	**.**	**.**	**T**		**…**	**…**	**…**	**…**	**…**	**.**	**.**	**.**	**.**
			RVA/Hu/AGO/CSMI10/2012/G8P[6]	**A**	**.**	**A**	**N**	**S**	**.**	**K**	**D**	**.**	**.**	**A**	**I**	**N**	**…**		**.**	**.**	**T**	**.**	**.**	**T**		**K**	**N**	**T**	**N**	**S**	**.**	**.**	**.**	**.**
			RVA/Hu/MWI/QEC287/2006/G8P[8]	**A**	**.**	**A**	**S**	**S**	**.**	**K**	**D**	**.**	**.**	**A**	**I**	**N**	**…**		**.**	**.**	**T**	**.**	**.**	**T**		**K**	**N**	**A**	**N**	**S**	**.**	**.**	**.**	**.**
	**I**		RVA/Hu/Croatia/CR2006/2006/G8P[8]	**T**	**.**	**A**	**S**	**S**	**.**	**K**	**E**	**.**	**.**	**A**	**I**	**N**	**K**		**.**	**.**	**T**	**.**	**.**	**T**		**K**	**S**	**A**	**N**	**S**	**.**	**.**	**.**	**.**
	**II**		RVA/Hu/ITA/PR1300/2004/G8P[14]	**V**	**.**	**A**	**S**	**S**	**.**	**K**	**D**	**.**	**.**	**A**	**I**	**N**	**K**		**.**	**.**	**T**	**.**	**.**	**T**		**K**	**N**	**A**	**N**	**S**	**.**	**.**	**.**	**.**
	**III**		RVA/Hu/AUS/BG8.01/1996/G8P[ND]	**V**	**.**	**A**	**S**	**S**	**.**	**K**	**D**	**.**	**.**	**A**	**I**	**N**	**K**		**.**	**.**	**T**	**.**	**.**	**T**		**K**	**N**	**A**	**N**	**S**	**.**	**.**	**.**	**.**
	**IV**		RVA/Hu/THA/TM-12505_60/2018/G8P[8]	**T**	**.**	**A**	**S**	**S**	**.**	**K**	**D**	**.**	**.**	**A**	**I**	**N**	**…**		**.**	**.**	**T**	**.**	**.**	**T**		**K**	**N**	**A**	**D**	**S**	**.**	**.**	**.**	**.**
**G9**	**VI**		**RVA/Hu/AGO/DV72HMCC/2022/G9P[ND]**	**T**	**.**	**G**	**T**	**E**	**.**	**K**	**.**	**.**	**.**	**A**	**I**	**.**	**…**		**.**	**N**	**T**	**A**	**D**	**.**		**K**	**D**	**S**	**T**	**.**	**.**	**.**	**S**	**.**
			RVA/Hu/VNM/VE17426/2020/G9P[8]	**T**	**.**	**G**	**T**	**E**	**.**	**K**	**.**	**.**	**.**	**A**	**I**	**.**	**K**		**.**	**N**	**T**	**A**	**D**	**.**		**K**	**D**	**S**	**T**	**.**	**.**	**.**	**S**	**.**
	**I**		RVA/Hu/USA/F45/1987/G9P[ND]	**A**	**.**	**G**	**T**	**E**	**.**	**K**	**D**	**.**	**.**	**A**	**I**	**.**	**K**		**.**	**N**	**T**	**A**	**D**	**T**		**K**	**D**	**S**	**T**	**.**	**.**	**.**	**S**	**.**
	**II**		RVA/Hu/IND/116E/1986/G9P[11]	**I**	**.**	**G**	**T**	**E**	**.**	**K**	**G**	**.**	**.**	**A**	**I**	**.**	**K**		**.**	**N**	**T**	**A**	**D**	**.**		**K**	**N**	**S**	**T**	**.**	**.**	**.**	**N**	**.**
	**III**		RVA/Hu/Malawi/MW47/1997/G9P[ND]	**T**	**.**	**G**	**T**	**E**	**.**	**K**	**D**	**.**	**.**	**A**	**I**	**.**	**K**		**.**	**N**	**T**	**A**	**D**	**.**		**K**	**D**	**S**	**T**	**.**	**.**	**.**	**S**	**.**
			RVA/Hu/AGO/C84/2012/G9P[6]	**T**	**.**	**G**	**T**	**E**	**.**	**K**	**D**	**.**	**.**	**A**	**I**	**.**	**…**		**.**	**N**	**T**	**A**	**D**	**.**		**K**	**D**	**S**	**T**	**.**	**.**	**.**	**S**	**.**
	**IV**		RVA/Hu/CHN/97’SZ37/1997/G9P[ND]	**T**	**.**	**G**	**T**	**E**	**.**	**K**	**D**	**.**	**.**	**A**	**I**	**.**	**K**		**.**	**N**	**T**	**A**	**D**	**.**		**K**	**D**	**S**	**T**	**.**	**.**	**.**	**S**	**.**
	**V**		RVA/Hu/USA/OM46/1997/G9P[8]	**T**	**.**	**G**	**T**	**E**	**.**	**K**	**D**	**.**	**.**	**A**	**I**	**.**	**K**		**.**	**N**	**T**	**A**	**D**	**.**		**K**	**D**	**S**	**T**	**.**	**.**	**.**	**S**	**.**

RVA = *R. alphagastroenteritidis.*

## Data Availability

All the sequences obtained in the present work are available in the GenBank repository under accession numbers PQ139230 to PQ139236 (VP4 gene sequences) and PQ139237 to PQ139243 (VP7 gene sequences). Any remaining data are available on request from the authors.

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
