# Peer review of "Rotavirus alphagastroenteritidis: Circulating Strains After the Introduction of the Rotavirus Vaccine (Rotarix®) in Luanda Province of Angola"

_viruses, 2025, doi:10.3390/v17060858_

Round 1
Reviewer 1 Report
Comments and Suggestions for Authors
The paper of Vita et al. stands as a continuation of the previous work recently published focusing this time on the molecular epidemiology of RVA strains in Luanda province.
However, I have some concerns that should be addressed by the Authors
- In abstract at line 39, you inserted the following sentence: “The RVA-antigen immunochromatographic test 3 (SD Bioline™, Abbott, USA) was performed and positive samples were genotyped.”
This is not correct because only 16.7% (121/723) of positive samples have been genotyped
- Results line 195: “thirteen (randomly chosen) were confirmed by Sanger sequencing method”. It seems from the phylogenetic analysis that the sequences were fourteen. Because have you chosen only 14 samples? Were the sequencing results in agreement with the PCR results?
- About genotype diversity, age and severity of disease, I suggest limiting your results only to the 121 genotyped samples and not insert anything about the severity that has been the focus of the previous paper.
- Results lines 252-254: “In general, the RVA sequences from Luanda Province subjected to phylogenetic analysis exhibit low diversity among them, as they cluster within the same lineages of the different genotypes for both VP4 and VP7”.
Seven sequences for VP4 and seven sequences for VP7 are not enough to draw any conclusion
- Discussion lines 443-444: “The eventual impact of the amino acid changes in antigen recognition by the immune 443 system is speculative”. I agree with this sentence and suggest deleting this part from both results and discussion sections. The number of sequences analyzed is too low.
Author Response
Dear Reviewer,
Please find below our point-by-point responses to your comments below. We have carefully addressed each point and made the corresponding changes to the manuscript where appropriate. All modifications in response to your suggestions are marked in blue in the revised manuscript for clarity.
Reviewer 1
The paper of Vita et al. stands as a continuation of the previous work recently published focusing this time on the molecular epidemiology of RVA strains in Luanda province.
However, I have some concerns that should be addressed by the Authors
1. In abstract at line 39, you inserted the following sentence: “The RVA-antigen immunochromatographic test (SD Bioline™, Abbott, USA) was performed and positive samples were genotyped.”
This is not correct because only 16.7% (121/723) of positive samples have been genotyped
Response: Thank you for your observation. We agree that the original phrasing could be misleading. To accurately reflect the methodology, we have revised the sentence in the abstract to: “The RVA-antigen immunochromatographic test (SD Bioline™, Abbott, USA) was performed, and a subset of 121 positive samples was genotyped.”
2. Results line 195: “thirteen (randomly chosen) were confirmed by Sanger sequencing method”. It seems from the phylogenetic analysis that the sequences were fourteen. Because have you chosen only 14 samples? Were the sequencing results in agreement with the PCR results?
Response: Thank you for your valuable observation. We acknowledge that the previous text was unclear. Although a larger number of samples were initially selected for sequencing, due to logistical constraints and variability in amplification success, high-quality sequences were obtained from only 10 samples. Specifically, four samples yielded sequences for both VP4 and VP7 genes (resulting in eight sequences), while six samples yielded sequences for only one gene each—three for VP4 and three for VP7—resulting in a total of 14 sequences. To clarify this in the manuscript, we have revised the text in the Abstract, in Materials and Methods and Results sections.
Update in the Abstract:
“Ten strains were randomly selected for further Sanger sequencing.”
Updated section 2.4 – Sequencing of RVA strains:
“Ten faecal samples were randomly selected for Sanger sequencing of both VP7 and VP4 gene amplicons, with the aim of covering the range of detected G and P genotypes, including non-typable strains. DNA sequencing was performed by STABVIDA Laboratories (Costa da Caparica, Portugal) using the corresponding first-round PCR primers. Complete sequences for both VP4 and VP7 genes were successfully obtained from 4 samples. In 3 samples, only the VP4 gene could be sequenced, while in the remaining 3, only the VP7 gene was recovered. These differences were due to variability in PCR amplification efficiency and sequence quality among the samples”.
Updated sentence in Results: “One hundred and twenty-one RVA-positive samples were genotyped, and sequencing confirmation was obtained for ten randomly selected samples, yielding a total of 14 sequences.”
3. About genotype diversity, age and severity of disease, I suggest limiting your results only to the 121 genotyped samples and not insert anything about the severity that has been the focus of the previous paper.
Response: We thank the reviewer for this thoughtful suggestion. While we acknowledge that disease severity was addressed in our previous publication, we believe that presenting the association between genotype distribution and severity in the subset of 121 genotyped samples adds valuable epidemiological insight that was not previously explored. This integrated analysis may be of particular interest to readers seeking to understand potential correlations between circulating genotypes and clinical outcomes. Importantly, we have ensured that all results related to severity are limited to the genotyped subset to avoid any overlap or misinterpretation.
4. Results lines 252-254: “In general, the RVA sequences from Luanda Province subjected to phylogenetic analysis exhibit low diversity among them, as they cluster within the same lineages of the different genotypes for both VP4 and VP7”.
Seven sequences for VP4 and seven sequences for VP7 are not enough to draw any conclusion
Response: We appreciate the reviewer’s comment and fully agree that the limited number of sequences represents a constraint on the robustness of phylogenetic conclusions. For this reason, we have clearly acknowledged this as a limitation in the Discussion section of the manuscript. Furthermore, we have revised the corresponding sentence in the Results to reflect the preliminary nature of the observation, avoiding overinterpretation. The revised sentence now reads: “Despite being based on a limited number of sequences, the RVA strains analyzed in Luanda Province showed low diversity, clustering within the same lineages of the respective genotypes for both VP4 and VP7.”
5. Discussion lines 443-444: “The eventual impact of the amino acid changes in antigen recognition by the immune system is speculative”. I agree with this sentence and suggest deleting this part from both results and discussion sections. The number of sequences analyzed is too low.
Response: We thank the reviewer for this thoughtful suggestion. We fully agree that the limited number of sequences restricts any firm conclusions regarding the immunological impact of the observed amino acid changes. However, we believe that a brief and explicitly cautious mention of this point is still relevant, as it highlights a potential direction for future investigation. To address the reviewer’s concern, we have carefully reworded the sentence to underscore its speculative nature, without drawing any strong conclusions.
The sentence now reads: “Given the limited number of sequences analyzed, any interpretation of amino acid changes remains speculative. Nonetheless, substitutions involving residues with differing chemical properties—such as nonpolar, polar, and charged amino acids—in key regions of VP4 (VP8) and VP7 were observed, which may suggest potential conformational alterations in antigenic epitopes. However, the biological relevance of these putative changes is uncertain and requires confirmation through analysis of a larger dataset.”
We hope this revised phrasing appropriately balances caution with scientific curiosity and provides useful context for readers.
Reviewer 2 Report
Comments and Suggestions for Authors
In this study, the authors observed genotype distribution of RVA strains in Luanda Province of Angola in 2021-2022. Overall, the manuscript is highly descriptive, and the biological significance of the observed genotype distribution was unclear. I do no think this manuscript is suitable for publication in Viruses. I would rather recommend the authors to report only the observed genotype distribution in some clinical journals.
- The BIC-selected models used for constructing ML trees should be presented.
- It was not clear how the chi-square test was conducted. The authors should clearly describe how it was done.
- The authors state that they sequenced 13 RVA strains, but show only 7 sequences each for VP4 and VP7. The authors should explain this discrepancy.
- The percentage identity analysis of VP4 and VP7 appears to be meaningless.
Author Response
Dear Reviewer,
Please find below our point-by-point responses to your comments below. We have carefully addressed each point and made the corresponding changes to the manuscript where appropriate. All modifications in response to your suggestions are marked in blue in the revised manuscript for clarity.
Reviewer 2
In this study, the authors observed genotype distribution of RVA strains in Luanda Province of Angola in 2021-2022. Overall, the manuscript is highly descriptive, and the biological significance of the observed genotype distribution was unclear. I do no think this manuscript is suitable for publication in Viruses. I would rather recommend the authors to report only the observed genotype distribution in some clinical journals.
- The BIC-selected models used for constructing ML trees should be presented.
Response: We thank the reviewer for this valuable suggestion. In response, we have now included the specific nucleotide substitution models used for constructing each phylogenetic tree, selected based on the lowest Bayesian Information Criterion (BIC) score. Specifically, the Hasegawa-Kishino-Yano model was used for the VP4 tree, and the Tamura 3-parameter model for the VP7 tree. This information has been added to the Materials and Methods section. The revised text (line 177) reads: “The nucleotide substitution model for each tree was selected based on the lowest Bayesian Information Criterion (BIC) score and was also used to calculate the pairwise identity percentages. For the VP4 phylogenetic tree, the Hasegawa-Kishino-Yano model was selected, while the Tamura 3-parameter model was used for the VP7 tree. The evolutionary history was inferred using the Maximum Likelihood method [24] with a Bootstrap test of 1000 replicates to assess reliability.”
2. It was not clear how the chi-square test was conducted. The authors should clearly describe how it was done.
Response: We appreciate the reviewer’s observation. To clarify the methodology, we have revised the relevant section to provide more detail on how the chi-square test was conducted. The updated text, now found at lines 191–199, reads:
“Descriptive statistics were used to assess the characteristics of children enrolled in this study. Data were summarized with number, and absolute (n) and relative (%) frequencies of categorical variables were calculated. To investigate a possible association between age or severity of disease and genotypes (G or P), children were divided into four age groups (0-6, 7-12, 13-24, > 24 months) and 3 AGE groups according to the Vesikari scoring system (mild, moderate, severe), respectively. The association between proportions was assessed with a chi-squared (χ2) test of independence. A p-value < 0.05 was set as the threshold to assess statistical significance. Data were analysed using the software SPSS version 22 (IBM Corp, Armonk, NY, USA).”
3.The authors state that they sequenced 13 RVA strains, but show only 7 sequences each for VP4 and VP7. The authors should explain this discrepancy.
Response: Thank you for your valuable observation. As indicated in the answer to reviewer 1 (see above) a larger number of samples were initially selected for sequencing, due to logistical constraints and variability in amplification success, high-quality sequences were obtained from only 10 samples. Specifically, four samples yielded sequences for both VP4 and VP7 genes (resulting in eight sequences), while six samples yielded sequences for only one gene each—three for VP4 and three for VP7—resulting in a total of 14 sequences. To clarify this in the manuscript, we have revised the text in the Abstract, Materials and Methods and Results sections.
Update in the Abstract (line 43):
“Ten strains were randomly selected for further Sanger sequencing.”
Updated section 2.4 – Sequencing of RVA strains (lines 159-166):
“Ten faecal samples were randomly selected for Sanger sequencing of both VP7 and VP4 gene amplicons, with the aim of covering the range of detected G and P genotypes, including non-typable strains. DNA sequencing was performed by STABVIDA Laboratories (Costa da Caparica, Portugal) using the corresponding first-round PCR primers. Complete sequences for both VP4 and VP7 genes were successfully obtained from 4 samples. In 3 samples, only the VP4 gene could be sequenced, while in the remaining 3, only the VP7 gene was recovered. These differences were due to variability in PCR amplification efficiency and sequence quality among the samples”.
Updated sentence in Results (lines 212-214): “One hundred and twenty-one RVA-positive samples were genotyped, and sequencing confirmation was obtained for ten randomly selected samples, yielding a total of 14 sequences.”
4. The percentage identity analysis of VP4 and VP7 appears to be meaningless.
Response: We thank the reviewer for this critical comment. To clarify, the percentage identity values reported for VP4 and VP7 were derived from pairwise nucleotide distance matrices generated using MEGA 7.0.23 with the p-distance model, as described in the Materials and Methods section. The identity percentages were calculated using the formula: identity = 1 – distance. These values were included to provide a general overview of the genetic variability within each gene segment among the studied strains, supporting the phylogenetic analyses. While we acknowledge that percentage identity is a descriptive measure, we believe it offers useful context when interpreting the diversity of circulating genotypes. To enhance transparency and reproducibility, the complete nucleotide distance matrices used for these calculations have now been included as Supplementary Figures S1 and S2. We hope this clarification addresses the reviewer’s concern and better conveys the intended purpose of this analysis.
Reviewer 3 Report
Comments and Suggestions for Authors
Dear Author:
Thank you for the opportunity to review your manuscript titled "Rotavirus alphagastroenteritidis circulating strains after the introduction of the rotavirus vaccine (Rotarix®) in Luanda Province of Angola."
After a careful and thorough evaluation, I commend you and your co-authors for conducting a well-structured and timely study that contributes valuable insights into post-vaccination rotavirus epidemiology in Angola. Your work addresses an important gap in surveillance data and provides a solid foundation for future research and public health policy.
I am pleased to inform you that I recommend the manuscript for acceptance with minor revisions. My comments focus on clarifying certain methodological aspects, enhancing the discussion, and addressing a few points related to language and presentation. These revisions are intended to strengthen the clarity and impact of your work.
Please find my detailed comments and suggestions below:
1. The introduction should provide some figures on the epidemiology of rotavirus in Angola, both before 2014 and at the present time.
2. Angola's rotavirus vaccine coverage ( Luanda Province and across the country). Angola has National Diroctorate of Public Health. What is the coverage accoding to the NDPH.
3. What rotavirus genotypes are circulating among the elderly in Angola? Are there any figures on this? As more young children (under 5 years old) are vaccinated, rotaviruses circulate more among unvaccinated adults, and from there, they can jump back to children. Has this situation been monitored in Angola?
4. Given that children born in 2014—the year Angola introduced the rotavirus vaccine—are now approximately 12 years old, could the authors clarify whether the study considered the proportion of rotavirus AGE cases among vaccinated cohorts? Specifically, was any analysis conducted to assess the incidence of rotavirus AGE among children born after 2014, in order to evaluate the long-term effectiveness of the vaccination program in Angola?
5. According to the introduction: the G1P[8] rotavirus genotype is predominant in North America, Europe, and Australia, accounting for over 70% of RVA infections. In contrast, its prevalence is significantly lower in South America (30%), Asia (30%), and Africa (23%). Given this distribution, why did health authorities in Angola opt to include Rotarix® in the national immunization program instead of RotaTeq®, which offers broader protection against multiple genotypes and could potentially prevent the spread of additional strains circulating in Angola? Furthermore, what are the most prevalent rotavirus genotypes currently identified in Angola?
6. The results section should begin with a comprehensive description of the study cohort. This includes demographic and clinical characteristics such as the distribution of participants by sex (male and female), age groups, and vaccination status. Additionally, it should detail the timing of symptom onset in relation to vaccination, providing insights into the temporal association between immunization and the development of acute gastroenteritis symptoms. This foundational information is essential for contextualizing the subsequent virological and genotypic findings.
7. In Section 3.1, the authors state that 13 samples were randomly selected from the 121 genotyped samples for Sanger sequencing. However, Section 3.3 presents analysis results for only 7 of these samples, with no information provided regarding the remaining 6. Could the authors please clarify the status of these 6 samples? Were they excluded due to quality issues, sequencing failure, or other reasons?
8. Language and Style: While the manuscript is generally well-written, there are occasional grammatical inconsistencies and awkward phrasings. A thorough language revision by a native English speaker or professional editor is recommended to improve clarity and flow.
Sincerely,
Reviewer
Author Response
Dear Reviewer,
Please find below our point-by-point responses to your comments below. We have carefully addressed each point and made the corresponding changes to the manuscript where appropriate. All modifications in response to your suggestions are marked in blue in the revised manuscript for clarity.
Reviewer 3
Thank you for the opportunity to review your manuscript titled "Rotavirus alphagastroenteritidis circulating strains after the introduction of the rotavirus vaccine (Rotarix®) in Luanda Province of Angola."
After a careful and thorough evaluation, I commend you and your co-authors for conducting a well-structured and timely study that contributes valuable insights into post-vaccination rotavirus epidemiology in Angola. Your work addresses an important gap in surveillance data and provides a solid foundation for future research and public health policy.
I am pleased to inform you that I recommend the manuscript for acceptance with minor revisions. My comments focus on clarifying certain methodological aspects, enhancing the discussion, and addressing a few points related to language and presentation. These revisions are intended to strengthen the clarity and impact of your work.
Please find my detailed comments and suggestions below:
- The introduction should provide some figures on the epidemiology of rotavirus in Angola, both before 2014 and at the present time.
Response: We thank the reviewer for this important comment. Prior to the introduction of the rotavirus vaccine (Rotarix) into Angola’s routine childhood immunization program in 2014, very limited data were available on rotavirus epidemiology in the country. To our knowledge, the only studies conducted before vaccine implementation were those carried out by our group during 2011–2012 (Esteves et al., 2016; https://doi.org/10.1002/jmv.24510) and 2012–2013 (Gasparinho et al., 2017; https://doi.org/10.1371/journal.pone.0176046), which were published in 2016 and 2017, respectively. Beyond these, the only additional data we were able to identify comes from a PATH report (https://media.path.org/documents/VAD_rotavirus_angola_fs.pdf), which cites WHO surveillance estimates from 2008, as reported in 2012 (World Health Organization. 2008 rotavirus deaths, under 5 years of age, as of 31 January 2012 [spreadsheet]. Available at: http://www.who.int/immunization/monitoring_surveillance/burden/estimates/rotavirus/en/index.html).
As for the current situation, rotavirus-specific diagnostics are not routinely performed in public hospitals in Angola. Diarrheal cases are managed according to WHO guidelines for paediatric diarrheal disease; however, laboratory confirmation of rotavirus infection is not part of standard clinical practice. General data on diarrheal disease burden and epidemiology are reported in the annual health reports published by the Ministry of Health of Angola.
We hope this information provides a clearer context for interpreting our study and highlights the challenges and importance of continued rotavirus surveillance in the region.
- Angola's rotavirus vaccine coverage (Luanda Province and across the country). Angola has National Directorate of Public Health. What is the coverage according to the NDPH.
Response: We thank the reviewer for this pertinent question. According to data from the Ministry of Health of Angola covering the period from 2014 to 2021, the national coverage for the rotavirus vaccine (Rotarix) in 2021 was 68% for the first dose and 55% for the second dose. In Luanda Province, coverage was slightly higher, with 74% for the first dose and 62% for the second dose. The WHO/UNICEF estimates for the same year indicated a lower overall coverage, reporting 36% as the estimated total coverage, although the officially reported figure in that report was 55%. This discrepancy may reflect differences in data sources or estimation methods. The decline in reported vaccine coverage from 2020 to 2021 has been attributed to the diversion of national health resources toward the COVID-19 response, as well as reported subnational disruptions in vaccine supply for several antigens.
The most recent estimate available from WHO for Rotarix coverage in Angola in 2023 is 46%, although it should be noted that this is a modelled estimate rather than a figure based on administrative coverage data. These data are accessible through the WHO/UNICEF Estimates of National Immunization Coverage (WUENIC) platform:
https://worldhealthorg.shinyapps.io/wuenic-trends/
We hope this information provides the necessary context regarding vaccine coverage in Angola and particularly in Luanda Province.
- What rotavirus genotypes are circulating among the elderly in Angola? Are there any figures on this? As more young children (under 5 years old) are vaccinated, rotaviruses circulate more among unvaccinated adults, and from there, they can jump back to children. Has this situation been monitored in Angola?
Response: We thank the reviewer for raising this important and timely issue. Unfortunately, to our knowledge, no studies have been conducted in Angola to investigate rotavirus infection or genotype circulation among the elderly or adult populations. Existing rotavirus surveillance and epidemiological research in the country has focused almost exclusively on children under five years of age, particularly in the context of vaccine introduction and impact assessment.
We were unable to identify any published data or official reports from the Ministry of Health or other sources that address rotavirus prevalence or genotype distribution in older age groups. As such, this represents an important knowledge gap in the country’s rotavirus surveillance landscape.
We agree that the dynamics of rotavirus transmission in partially vaccinated populations—including potential shifts toward increased circulation among unvaccinated adults—are highly relevant and warrant further investigation. We hope future studies can address this issue to better inform public health strategies in Angola and similar settings.
- Given that children born in 2014—the year Angola introduced the rotavirus vaccine—are now approximately 12 years old, could the authors clarify whether the study considered the proportion of rotavirus AGE cases among vaccinated cohorts? Specifically, was any analysis conducted to assess the incidence of rotavirus AGE among children born after 2014, in order to evaluate the long-term effectiveness of the vaccination program in Angola?
Response: We thank the reviewer for this thoughtful and relevant question. Our study was designed as a cross-sectional, hospital-based investigation aimed at estimating the detection rate of rotavirus infection and characterizing the circulating strains among children under five years of age, conducted in six hospitals across Luanda Province during the period 2021–2022, which falls within the post-vaccination era. Given the study design and target age group, we did not conduct cohort-based analyses, nor did we assess incidence trends specifically among vaccinated versus unvaccinated cohorts. To our knowledge, no longitudinal or cohort studies evaluating the long-term effectiveness of the rotavirus vaccination program in Angola—particularly among children born after 2014—have been conducted or published to date.
We agree that such analyses would provide valuable insights into vaccine impact and the evolution of rotavirus epidemiology in the country. We believe this is an important area for future research and encourage its prioritization as part of Angola’s ongoing efforts to monitor and strengthen its immunization program.
- According to the introduction: the G1P[8] rotavirus genotype is predominant in North America, Europe, and Australia, accounting for over 70% of RVA infections. In contrast, its prevalence is significantly lower in South America (30%), Asia (30%), and Africa (23%). Given this distribution, why did health authorities in Angola opt to include Rotarix® in the national immunization program instead of RotaTeq®, which offers broader protection against multiple genotypes and could potentially prevent the spread of additional strains circulating in Angola? Furthermore, what are the most prevalent rotavirus genotypes currently identified in Angola?
Response: We thank the reviewer for this insightful question. The genotype distribution figures cited in the Introduction are based on more recent studies that reflect shifts in genotype prevalence over time. However, the decision to introduce Rotarix® into Angola’s national immunization program was made in 2014, based on the best available evidence at that time. Earlier data from the African Rotavirus Surveillance Network—as reported by Mwenda et al. (2014; doi:10.1097/INF.0000000000000174)—indicated that G1P[8] was the most prevalent genotype across the African continent during the pre-vaccine period. This was consistent with findings from our own pre-vaccination studies in Angola (Esteves et al., 2016; https://doi.org/10.1002/jmv.24510), which also identified G1P[8] as the predominant strain. Therefore, the selection of Rotarix®, which targets the G1P[8] genotype, was aligned with both national and regional genotype data at the time. Additional factors influencing the choice included the logistical advantages of a two-dose schedule, which simplifies implementation in resource-limited settings, and the fact that Angola was eligible for support through Gavi, the Vaccine Alliance, which had already negotiated procurement of Rotarix® for several low-income countries.
Regarding currently circulating genotypes, our study presents the most recent data available for Angola, a country where surveillance remains limited. The genotypes identified in our analysis represent the latest known circulating strains, and we hope our findings contribute to the broader understanding of rotavirus epidemiology in post-vaccine settings.
- The results section should begin with a comprehensive description of the study cohort.This includes demographic and clinical characteristics such as the distribution of participants by sex (male and female), age groups, and vaccination status. Additionally, it should detail the timing of symptom onset in relation to vaccination, providing insights into the temporal association between immunization and the development of acute gastroenteritis symptoms. This foundational information is essential for contextualizing the subsequent virological and genotypic findings.
Response: We thank the reviewer for this valuable suggestion. The detailed demographic and clinical characteristics of the study cohort—including distribution by sex, age groups, vaccination status, and timing of symptom onset—were previously published in a companion article directly related to this study (Vita et al., 2024; https://doi.org/10.3390/v16121949). This publication is already cited in the manuscript (Reference [17]).
- In Section 3.1, the authors state that 13 samples were randomly selected from the 121 genotyped samples for Sanger sequencing. However, Section 3.3 presents analysis results for only 7 of these samples, with no information provided regarding the remaining 6. Could the authors please clarify the status of these 6 samples? Were they excluded due to quality issues, sequencing failure, or other reasons?
Response: Thank you for your valuable observation. As indicated in the answer to reviewers 1 and 2 (see above) a larger number of samples were initially selected for sequencing, due to logistical constraints and variability in amplification success, high-quality sequences were obtained from only 10 samples. Specifically, four samples yielded sequences for both VP4 and VP7 genes (resulting in eight sequences), while six samples yielded sequences for only one gene each—three for VP4 and three for VP7—resulting in a total of 14 sequences. To clarify this in the manuscript, we have revised the text in the Abstract, Materials and Methods and Results sections.
Update in the Abstract (line 43):
“Ten strains were randomly selected for further Sanger sequencing.”
Updated section 2.4 – Sequencing of RVA strains (lines 159-166):
“Ten faecal samples were randomly selected for Sanger sequencing of both VP7 and VP4 gene amplicons, with the aim of covering the range of detected G and P genotypes, including non-typable strains. DNA sequencing was performed by STABVIDA Laboratories (Costa da Caparica, Portugal) using the corresponding first-round PCR primers. Complete sequences for both VP4 and VP7 genes were successfully obtained from 4 samples. In 3 samples, only the VP4 gene could be sequenced, while in the remaining 3, only the VP7 gene was recovered. These differences were due to variability in PCR amplification efficiency and sequence quality among the samples”.
Updated sentence in Results (lines 212-214): “One hundred and twenty-one RVA-positive samples were genotyped, and sequencing confirmation was obtained for ten randomly selected samples, yielding a total of 14 sequences.”
- Language and Style: While the manuscript is generally well-written, there are occasional grammatical inconsistencies and awkward phrasings. A thorough language revision by a native English speaker or professional editoris recommended to improve clarity and flow.
Response: Thank you for this suggestion. We have thoroughly revised the English language throughout the manuscript, and we hope that the clarity and overall quality of the text have now been improved. The changes made during the language revision process are marked in blue. We trust that the current version meets the language and editorial standards required for publication.
Round 2
Reviewer 1 Report
Comments and Suggestions for Authors
No further comments
Author Response
Dear Academic Editor,
We thank you very much for your comments.
Please find below our point-by-point responses to your comments. We have carefully addressed each point you raised and made the corresponding changes into the manuscript.
All modifications in response to your suggestions are marked in red in the revised manuscript for clarity.
- p. 1, line 43: ten strains must be ten samples or isolates.
Response: Thank you for your observation. As suggested, at line 43 we corrected as follows:
“Ten samples were randomly selected for further Sanger sequencing”.
- Rotavirus alphagastroenteritidis is a virus species name which usage is subject to strict rules. Accordingly, it may be abbreviated to R. alphagastroenteritidis (as common in biology) but not RVA (as was common in studies of rotaviruses). Correct it across the manuscript.
Response: Thank you for this suggestion. We corrected throughout the test “RVA” to “R. alphagastroenteritidis”.
- Detail and discuss any evidence or its lack that the genetic makeup of the Rotarix vaccine may have affected genotype composition of the currently circulating R. alphagastroenteritidis."
Response: Thank you very much for this very important suggestion. To better discuss this issue, we added a new sentence at line 454 and highlighted in red
…”It is likely that the introduction of the vaccine or natural temporal fluctuations created selective pressure, driving the observed change in the circulating genotypes, phenomenon reported in other geographic regions”.
We hope that this final version meets the editorial standards required for publication.
With my best regards,
Claudia Istrate (DVM, PhD)